# Spatiotemporal Land Use Change Detection Through Automated Sampling and Multi-Feature Composite Analysis: A Case Study of the Ebinur Lake Basin

**DOI:** 10.3390/s25144314

**Published:** 2025-07-10

**Authors:** Yi Yang, Liang Zhao, Ya Guo, Shihua Liu, Xiang Qin, Yixiao Li, Xiaoqiong Jiang

**Affiliations:** 1China Aero Geophysical Survey and Remote Center for Natural Resources, Beijing 100083, China; 15652940083@163.com (Y.Y.); gy939251@163.com (Y.G.); 18513558076@163.com (S.L.); qinxiangcags@yeah.net (X.Q.); leeshell199801@gmail.com (Y.L.); jiangxq17@163.com (X.J.); 2Institute of Remote Sensing Satellite, China Academy of Space Technology, Beijing 100094, China

**Keywords:** sample automation, feature optimization, multi-feature composite, image classification, land use change, remote sensing, machine learning

## Abstract

Land use change plays a pivotal role in understanding surface processes and environmental dynamics, exerting considerable influence on regional ecosystem management. Traditional monitoring approaches, which often rely on manual sampling and single spectral features, exhibit limitations in efficiency and accuracy. This study proposes an innovative technical framework that integrates automated sample generation, multi-feature optimization, and classification model refinement to enhance the accuracy of land use classification and enable detailed spatiotemporal analysis in the Ebinur Lake Basin. By integrating Landsat data with multi-temporal European Space Agency (ESA) products, we acquired 14,000 pixels of 2021 land use samples, with multi-temporal spectral features enabling robust sample transfer to 12028 pixels in 2011 and 10,997 pixels in 2001. Multi-temporal composite data were reorganized and reconstructed to form annual and monthly feature spaces that integrate spectral bands, indices, terrain, and texture information. Feature selection based on the Gini coefficient and Out-Of-Bag Error (OOBE) reduced the original 48 features to 23. In addition, an object-oriented Gradient Boosting Decision Tree (GBDT) model was employed to perform accurate land use classification. A systematic evaluation confirmed the effectiveness of the proposed framework, achieving an overall accuracy of 93.17% and a Kappa coefficient of 92.03%, while significantly reducing noise in the classification maps. Based on land use classification results from three different periods, the spatial distribution and pattern changes of major land use types in the region over the past two decades were investigated through analyses of ellipses, centroid shifts, area changes, and transition matrices. This automated framework effectively enhances automation, offering technical support for accurate large-area land use classification.

## 1. Introduction

Global climate change and intensified human activities have profoundly affected the Earth’s surface. As a key component of global environmental change, land use change directly impacts ecosystem functions and regional sustainability [1]. Therefore, accurately and promptly monitoring and analyzing the spatiotemporal dynamics of land use is essential for understanding regional environmental evolution, developing effective land management strategies, and addressing future environmental challenges. Remote sensing technology, characterized by its wide spatial coverage, rapid data acquisition, and high temporal resolution, has become a fundamental tool for land use monitoring and related research [2].

Multi-source remote sensing data, including multispectral, hyperspectral, thermal infrared, radar satellite, and UAV data, are widely used for land use classification and monitoring [3]. Each data source has unique strengths, and some are complementary. Consequently, extensive research efforts have been devoted to leveraging multi-source data fusion for more effective land use classification [4]. For example, Shuang Shuai et al. classified two vegetation cover areas in Haixi Prefecture, Qinghai Province, using Gaofen and Sentinel data, demonstrating that integrating different feature sets enhanced classification accuracy [5]. Filippi and Jensen used AVIRIS hyperspectral data and neural networks to accurately identify coastal vegetation [6]. Nevertheless, for monitoring land use changes, the Landsat series data remain the preferred choice for many researchers due to their extensive temporal coverage, offering continuous imagery spanning several decades. This characteristic renders the data particularly valuable for long-term land use change monitoring and analysis [7].

High-quality land use samples and optimal feature selection are crucial for accurate land use mapping [8]. Traditional methods often rely on manual sample selection. For example, Runxiang Li et al. used field GPS to collect samples and explored the scale effect of image spatial resolution on land cover classification using multi-sensor data [9]. Hao Yu et al. combined Sentinel-1/2 imagery, terrain data, and field samples from handheld BeiDou and drones to map wetland land use [10]. However, manual sampling is time-consuming, labor-intensive, and prone to human bias, hindering large-scale, long-term monitoring. Recent research increasingly focuses on automated sample generation [11]. Yanglin Cui et al. used Landsat 8 focal statistics and phenology to determine optimal focal radii for different land use types, and automatically generated training samples, achieving 92.99% accuracy with Random Forest [12]. Yanan Wen proposed an automated methodology for collecting stable and representative maize samples from the Cropland Data Layer (CDL) product. Utilizing adaptively selected training samples, this approach enhanced maize mapping performance at large scales and in early stages, achieving an average overall accuracy of approximately 88% [13]. Relying on a single spectral feature often inadequately captures the spectral heterogeneity inherent in complex land cover types, thus limiting classification accuracy and robustness [4]. Consequently, the increasing incorporation of multiple features in classification studies introduces challenges such as feature interference and redundancy. Optimizing these feature sets is, therefore, a critical issue requiring resolution [14]. Yi Zhou et al. used out-of-bag data with recursive feature elimination for optimization, achieving 93.64% overall accuracy and a 92.60% Kappa with Random Forest [15]. Jinxi Yao et al. used the Gini index to rank feature importance and selected features based on overall accuracy and Kappa coefficient relationships [16]. The Gini coefficient is widely used for variable importance assessment and feature optimization due to its ease of calculation, interpretability, and broad applicability [14,15,16].

The classifier selection is crucial for accurate land use type identification. Classification models broadly include traditional machine learning algorithms like Support Vector Machine, Maximum Likelihood Classification, Random Forest, K-Nearest Neighbor, Recursive Boosting Tree, and Object-Oriented Classification, as well as deep learning models such as Deep Neural Networks, Recurrent Neural Networks, and Convolutional Neural Networks [17]. Giancarlo Alciaturi et al. mapped summer and winter land use in Uruguay using Random Forest, Support Vector Machine, and Gradient Boosting Tree classifiers. Their findings confirmed the superior performance of Random Forest and Gradient Boosting Tree [18]. Chen Zhang et al. mapped forest land using Sentinel-2 imagery and classifiers including Support Vector Machine, K-Nearest Neighbor, Random Forest, Decision Tree, and Multilayer Perceptron, achieving over 95% accuracy [19]. Pongana et al. compared RF, Classification and Regression Trees, and Gradient Boosting Tree for rice paddy extraction in Thailand and Laos using superpixel segmentation based on a simple non-iterative clustering algorithm [20]. Muhammad Fayaz et al. investigated transfer learning with Inception-v3 and DenseNet121 architectures to develop a reliable land area classification system for urban land use identification. Their results demonstrated the method’s effectiveness and yielded significant results [21]. Qichi Yang et al. proposed an alpine land cover classification method based on Deep Convolutional Neural Networks and multi-source remote sensing data, achieving 86.24% overall accuracy and a Kappa coefficient of 81.56% [22]. Fan et al. proposed a Hierarchical Convolutional Recurrent Neural Network classification model. This model combines CNN and RNN modules for pixel-level land cover classification using multispectral remote sensing data. They reported an overall accuracy of 97.62% on the Sentinel-2 dataset [23]. While deep learning models can offer higher accuracy in certain contexts, they demand large, high-quality datasets, complex training, and significant time for large-area studies [24]. Prior research suggests RF, SVM, and object-based classification are excellent methods for large-area land use mapping [18,19,20].

Based on the aforementioned research, this study aims to develop a highly automated technical framework that enables automated sample generation, the optimization of multidimensional feature spaces, and the construction of classification models. By utilizing multi-temporal ESA classification products and long-term Landsat imagery, multi-temporal composite data were constructed through data reorganization and reconstruction analysis. Multiple classification models were then developed to achieve accurate land use classification and spatiotemporal change analysis in the Ebinur Lake Basin.

## 2. Study Area and Data Source

### 2.1. Study Area

The Ebinur Lake Basin is geographically situated in the northwestern part of Xinjiang, China, on the southwestern fringe of the Junggar Basin. It is located deep within the Eurasian continent, spanning between 43°39′ and 45°53′ N latitude, and 79°54′ and 85°16′ E longitude, encompassing an area of approximately 48,561 square kilometers [25]. The administrative region primarily covers Wenquan County, Bole City, Shuanghe City, Alashankou City, Jinghe County, and the western part of Usu City (Figure 1). This basin serves as a crucial node along the “Belt and Road” initiative and stands as a significant grain, cotton, and livestock production base in Northern Xinjiang [26].

The study area is characterized by a typical temperate continental arid climate, with elevations ranging from 187 to 4133 m above sea level. The region experiences high evapotranspiration rates, with a mean annual temperature of 5.6 °C. Precipitation is scarce, concentrated primarily in the mountainous areas during the summer months, with an annual average of approximately 200 mm. In contrast, the mean annual potential evapotranspiration exceeds 1500 mm [27]. The primary inflow to Lake Ebinur originates from three rivers: the Bortala River, the Jing River, and the Kuytun River. However, in recent decades, extensive agricultural expansion has resulted in significant anthropogenic water diversion for irrigation, which has increasingly exacerbated ecological and environmental issues. Therefore, conducting accurate spatiotemporal analyses of land use changes is essential for effectively addressing these emerging challenges [28].

### 2.2. Satellite Data Processing

In previous studies, the Sentinel and Landsat satellite series have been widely used for land use classification, feature extraction, and the spatiotemporal analysis of land cover types. Sentinel data, with a spatial resolution of 10 m and a dual-satellite revisit cycle of six days, are particularly suitable for high-precision, single-date land use mapping. Although Landsat data offer a coarser spatial resolution of 30 m compared to Sentinel data, they provide the distinct advantage of long-term temporal coverage, making them valuable for historical and time-series analyses [29]. The availability of Landsat data dating back to 1984 within the Google Earth Engine (GEE) platform renders them particularly conducive to the long-term spatiotemporal change analysis of land use types [30].

The study utilizes the GEE platform to retrieve and process Landsat series images, including Landsat-5, Landsat-7, and Landsat-8. GEE is an open-access platform developed through a collaboration among Google, Carnegie Mellon University, and the United States Geological Survey (USGS). Its key advantages include direct access to preprocessed satellite imagery, efficient computation for pixel and time-series-based analyses, and the seamless integration of machine learning algorithms for tasks such as feature extraction, thereby enhancing Earth observation capabilities (https://code.earthengine.google.com/, accessed on 15 February 2025) [31].

To ensure the quality of remote sensing data for regional land use type classification and spatiotemporal change analysis, preprocessed Landsat-5, Landsat-7, and Landsat-8 data were retrieved from GEE using the datasets “LANDSAT/LT05/C02/T1_L2”, “LANDSAT/LE07/C02/T1_L2”, and “LANDSAT/LC08/C02/T1_L2”, respectively. Multispectral Landsat imagery from the years 2001, 2011, and 2021 covering the Ebinur Lake Basin was selected as the primary remote sensing data source (Table 1). Leveraging the high-performance analytical and computational capabilities of the GEE platform, the remote sensing data were further processed online, including band selection, cloud removal, and image cropping, to obtain high-quality datasets for subsequent analysis.

## 3. Method

The detailed research workflow is illustrated in Figure 2 and consists of the following main steps: (i) preprocessing of remote sensing image data; (ii) automatic generation and transfer of classification samples; (iii) multi-temporal data composite; (iv) construction of feature space and classification models; and (v) spatiotemporal analysis of land use type changes.

### 3.1. Samples Generation and Migration

By integrating historical remote sensing imagery data with existing surface feature information, the land use types within the Ebinur Lake Basin was classified into into seven categories: Forest, Grassland, Cropland, Artificial, Bare Land, Water Body and Others [32]. Considering factors such as the spatial resolution of the data, their accuracy, and the land use type classification information provided by the classification product, this study employed the ESA WorldCover classification product and Landsat-7 imagery to achieve the automatic generation of land use type samples for the Ebinur Lake Basin in 2021 and the migration of samples from 2011 and 2001 [33].

#### 3.1.1. Generating Land Use Samples for 2021, 2011, and 2001

Based on the existing classification products and definitions of land use types, further processing is applied to the ESA classification products for 2020 and 2021 to ensure a superior, high-quality sample dataset. In GEE, the “remap” function is used to convert the ESA classification system into the classification system adopted in this study, and the “updateMask” function is employed to obtain the pixels where the land use types remain unchanged between the two time periods. Building on this, a spatial filter is applied to process the pixels and select a central pixel that exhibits the same land use type characteristics as its eight surrounding pixels. Based on the filtered pixels, 2000 random sample points were generated for each land use category for 2021 [34].

Based on the 2021 land use sample data from the Ebinur Lake Basin and in combination with Landsat-7 remote sensing imagery from 2021, 2011, and 2001, this study calculates the spectral angle mapper and squared Euclidean distance between the land use type samples of the 2021 and 2011 images, as well as between the 2011 and 2001 images. By using these two indicators to measure the similarity of sample data across different years, unchanged samples can be identified and selected. These unchanged samples can then be used for classification tasks in the target year, thereby accomplishing the task of interannual transfer. The spectral angle and Euclidean distance thresholds were set to 0.4 and 0.5, respectively, to enable the migration of the 2021 samples to 2011 and the 2011 samples to 2001 [35].

As shown in Table 2, the automated method enabled the efficient generation of large sample datasets for the Ebinur Lake Basin across three time periods: 2021, 2011, and 2001. Specifically, 14,000, 12,028, and 10,997 pixels were obtained for each respective year, substantially reducing the manual annotation workload and improving both the efficiency and consistency of the sampling process. Among them, Forest, Grassland, and Cropland categories constituted the predominant sample proportions in each period, which aligns with the region’s characteristic reliance on agricultural and pastoral land use. The quantity of land use type samples produced for each time period is sufficient to satisfy the sample size requirements for land use remote sensing classification.

By effectively integrating the extraction of pixels with unchanged land use types from dual-temporal ESA data with spatial filtering techniques, we achieved a reduction in the spatial heterogeneity and pixel edge effects inherent in ESA data. This approach also mitigated the impact of geolocation errors and variable cross-layer spatial resolution [34]. As evidenced by the sample distribution maps for three periods within the Ebinur Lake Basin (Figure 3), the samples exhibited comprehensive coverage across the entire study area, with a relatively uniform distribution. Each land use type was adequately sampled within its respective dominant distribution areas, thereby enhancing data representativeness. This improved data quality provides robust support for subsequent classification model training and accuracy assessment, contributing to enhanced model generalizability.

#### 3.1.2. Assessing the Quality of Land Use Samples

Objectively, the sample quality demonstrated satisfactory performance in terms of both sample size and distribution. In order to quantify the generation and migration quality of samples among different land-use types, the separability is evaluated by calculating the Jeffries–Matusita (JM) distance metric [36]. The JM distance quantifies the separability of land cover classes by evaluating the statistical distance between their sample distributions, providing a robust and widely applicable metric for assessing inter-class differentiation. The resulting JM distance values range from 0 to 2, where larger values indicate greater separability. Specifically, a value greater than 1.8 signifies excellent sample separability, values between 1.4 and 1.8 indicate acceptable sample selection, and values below 1.4 imply poor separability among the classes [37]. The calculation process and formula for the JM distance are as follows:(1)$$JM=2\times (1-e^{-B})$$
where *B* represents the Bhattacharyya distance on a specific feature dimension. For the samples selected from different land cover types, the formula for calculating the Bhattacharyya distance *B* is given as follows:(2)$$B=\frac{1}{8}{\left(m_{1}-m_{2}\right)}^{2}\frac{2}{\delta_{1}^{2}+\delta_{2}^{2}}+\frac{1}{2}\ln\left[\frac{\delta_{1}^{2}+\delta_{2}^{2}}{2\delta_{1}\delta_{2}}\right]$$
where *m* represents the mean value of the land cover feature, and *δ* represents the variance of a specific land cover feature.

To evaluate the separability of samples across different years and feature sets, the feature space was further partitioned based on annual sample datasets. The JM distance was calculated between land cover classes to generate box plots illustrating the sample quality distribution (Figure 4). Overall, the 2021 sample set exhibited the highest mean JM distance across all feature types, with a relatively reasonable range of variation. This suggests that the land use samples from this period demonstrated more significant differences in classification features, indicating the best overall sample quality. Furthermore, Figure 4 reveals both differences and similarities in the JM distance distribution across the three periods under various feature types. Specifically, in terms of land use sample separability, vegetation index features exhibited the best performance, with a mean JM distance exceeding 1.90, followed by spectral band features, which achieved a mean JM distance above 1.20. Texture and terrain features exhibited relatively poorer performance, with mean JM distances distributed around 1.10. Notably, while vegetation index features demonstrated the highest overall performance, individual land use samples showed poorer JM values when used alone. However, this phenomenon was mitigated to some extent when utilizing the complete feature set, resulting in a mean JM distance above 1.96.

### 3.2. Composite Feature Construction

Considering that the Ebinur Lake Basin comprises vegetation land use types with significant temporal variations and non-vegetation types with relatively weak image changes, the study integrates the annual cycle with periods of vegetation growth differences to extract effective features for land use classification.

#### 3.2.1. Time Range Selection

To obtain comprehensive and high-quality remote sensing imagery of the Ebinur Lake Basin, we utilized the “merge” function on the GEE platform to reorganize Soil-Adjusted Vegetation Index (SAVI) data derived from multi-source Landsat imagery throughout the year. SAVI is known for its robustness in vegetation monitoring [38]. Furthermore, an annual SAVI dataset was synthesized at 10-day intervals.

To address issues such as outliers, invalid values, and noise in the data, the Savitzky–Golay (S-G) filtering method was employed to reconstruct the vegetation index data [39]; the statistical results of the SAVI time-series reconstruction effect using the proposed filtering method are shown in Table 3. The standard deviation, covariance, and correlation coefficient reflect the degree of approximation between the reconstructed and original curves. Lower standard deviation and covariance values, coupled with higher correlation coefficients, indicate superior curve fidelity [40]. For the S–G filtered results, the mean values for forest, grassland, and cropland were 0.13848, 0.15729, and 0.26862, respectively, with corresponding standard deviations of 0.06613, 0.09618, and 0.20057. The root mean square error and correlation coefficient between the original and fitted data were 0.01178 and 0.98433 for Forest, 0.01372 and 0.99046 for Grassland, and 0.02642 and 0.99181 for Cropland. The fitting statistics for these three vegetation types demonstrate that the S–G filter effectively removes noise and provides a superior fit to the original SAVI data.

Temporal analysis of the reconstructed data revealed a consistent ascending–descending pattern across the three vegetation classes, corresponding to the cyclical phenological progression from vegetative growth to reproductive maturity and, finally, to senescence. This periodic behavior provides a key temporal indicator for delineating phenological phases, which can significantly improve the precision of land use classification.

By analyzing the reconstructed data for Forest, Grassland, and Cropland, the intersection points of the three datasets were identified to select time periods with significant vegetation differences. The temporal filtering principle was implemented to maintain consistent SAVI value rankings among the three vegetation types throughout the selected period. Therefore, we identified the initial and final critical dates. In Figure 5, point “a” represents the initial critical value, and point “b” represents the final critical value. Within the temporal range defined by “a” and “b”, the SAVI values consistently followed the pattern: Cropland > Grassland > Forest.

#### 3.2.2. Classification Features Selection

Based on the annual composite data and the composite data for the selected time period, a feature space is constructed using bands, vegetation indices, textures, and terrain. The feature set includes (1) band features derived from Landsat data; (2) vegetation indices calculated from the band features as specified in Table 4; and (3) terrain features (aspect, slope, and hillshade) derived from the 30 m resolution SRTM DEM via standard geospatial analysis. In addition, texture features are extracted based on the principle of the gray-level co-occurrence matrix (GLCM) [41]. According to previous studies, the NIR, Red, and Green bands of the imagery are selected, and a weighted linear combination is used to obtain a grayscale image, which serves as the GLCM input image (Gray) for calculating the texture information of the remote sensing imagery [42]. The specific calculation formula is as follows:(3)Gray=0.3×NIR+0.59×Red+0.11×Green
where *NIR* represents the reflectance of the Near-Infrared band, *Green* represents the reflectance of the Green band, and *Red* represents the reflectance of the Red band.

### 3.3. Feature Space Optimization

Ideally, a greater number of variables can better represent the characteristics of land use types, thus improving classification accuracy. However, previous studies have shown that high feature correlation and increased computational complexity can negatively impact classification accuracy and efficiency [16]. To reduce computational complexity, the study utilizes Random Forest feature importance and OOBE for feature selection [15].

The Gini index measures the contribution of a feature to the classification result. A higher Gini index indicates a greater impact of the feature on the classification result, signifying higher classification ability [43]. Therefore, the Gini index is used to evaluate feature importance, and its specific calculation formula is as follows:

Given m features *X*_1_, *X*_2_, *…*, *X_m_*, the formula for calculating the Gini coefficient is(4)$${GI}_{m}=\sum_{k=1}^{\left|K\right|}\sum_{k^{’}\neq k}p_{mk}p_{mk}^{’}=1-\sum_{k=1}^{\left|K\right|}p_{mk}^{2}$$
where *K* represents the number of classes, and *p_mk_* represents the proportion of class *k* in node m.

Assuming the number of trees in the random forest is *n*, the formula for calculating the feature importance score of feature *X_j_* is(5)$${VIM}_{j}^{Gini}=\sum_{i=1}^{n}\sum_{m\in M}{VIM}_{jm}^{Gini}={GI}_{m}-{GI}_{l}-{GI}_{r}$$
where set *M* is the nodes in Decision Tree *i* where feature *X_j_* appears, *GI_m_* is the Gini coefficient of the node before branching, and *GI_l_* and *GI_r_* represent the Gini coefficients of the two nodes after branching, respectively.

Out-of-Bag (OOB) refers to the samples that are not included in the training set of a Decision Tree when using bootstrap sampling (random sampling with replacement) to generate the tree within a Random Forest model. Approximately 1/3 of the samples are left out. These unselected samples are called OOB samples. The OOBE, derived from these OOB samples, can be used to evaluate feature importance [44]. The formula for calculating OOBE is as follows:(6)$$OOBE=\frac{1}{n}\sum_{1}^{n}{\left(y_{i}-y_{i}^{OOB}\right)}^{2}$$
where $y_{i}$ represents the actual value of the dependent variable in the OOB data, and $y_{i}^{OOB}$ represents the predicted value obtained from the Random Forest model.

The steps for optimizing feature space based on the Gini coefficient and OOBE are as follows:By calculating the Gini coefficient score for each feature, obtain the importance sequence of each feature.Based on the feature importance ranking, sequentially select the top n most important features to construct N distinct feature subsets. Subsequently, train a Random Forest model on each of these subsets and obtain the OOBE for each model.Compare the OOBE of all models, and the feature subset used by the model with the smallest OOBE is the optimal feature set.

### 3.4. Classification Method

In the context of selecting land use type classification models for the Ebinur Lake Basin, the study built upon prior work by utilizing established algorithms, namely, Random Forest [45], Support Vector Machine [46], and Gradient Boosting Decision Tree [47], alongside a proposed object-oriented Gradient Boosting Decision Tree classification model [47,48]. The sample dataset was partitioned into training and validation subsets with an 8:2 split [48], and the optimized feature set was subsequently fed into each classifier to accomplish high-accuracy land use type classification.

#### 3.4.1. Random Forest (RF)

Random Forest is an ensemble learning algorithm that integrates multiple Decision Trees. It employs a bagging method to generate independent training sample sets for each decision tree, and the final classification is determined by a majority vote of all decision trees. This algorithm is effective in handling large-scale datasets and exhibits strong generalization ability, high robustness, and fast classification speed. Hyperparameter tuning is crucial for optimizing the performance of Random Forest [45]. Through multiple experiments, the number of Decision Trees is set to 30.

#### 3.4.2. Support Vector Machine (SVM)

The Support Vector Machine is a machine learning algorithm commonly used for classification and regression. In SVM classification, the algorithm attempts to find a hyperplane in the feature space that can separate the data into different classes with the maximum margin, where the margin is defined as the distance between the hyperplane and the nearest data point of each class. SVM classification offers advantages over other machine learning algorithms, such as high accuracy, the ability to handle high-dimensional data, and the capability to deal with non-linearly separable data. However, it can be computationally expensive and may require tuning of hyperparameters, such as the kernel function and regularization parameters [46]. In this study, the Radial Basis Function (RBF) kernel is used.

#### 3.4.3. Gradient Boosting Decision Tree (GTDB)

Gradient Boosting Decision Trees are an iterative algorithm in ensemble learning, an improvement based on boosting methods. By iteratively adding multiple weak classifiers, GBDT gradually learns complex patterns in the data. Each iteration finely optimizes the loss function associated with the model, thereby exploring better model parameter configurations. Parameter settings greatly influence the classification performance of Gradient Boosting Decision Trees [47]. In this study, the number of Decision Trees was set to 30.

#### 3.4.4. Object-Oriented Gradient Boosting Decision Tree (OO-GBDT)

The Object-Oriented Gradient Boosting Decision Tree is a classification model that effectively combines simple non-iterative clustering for superpixel segmentation and Gradient Boosting Decision Trees. It uses the Simple Non-Iterative Clustering algorithm to segment images, generating a set of land use objects (clusters) within the image [49]. Finally, it performs Gradient Boosting Decision Tree classification on these cluster units. This method can effectively improve clustering efficiency and classification performance and provide better smoothing effects for land use type classification boundaries, and it has obvious noise suppression effects.

The implementation of OO-GBDT is based on the “SNIC” function and “smileGradientTreeBoost” function in the GEE platform. The SNIC function sets the size, neighborhoodSize, compactness, connectivity, and seeds parameters as follows: The initial clustering center interval is 2 pixels, the neighborhood range is 10 pixels, spatial weights are not considered, the connectivity is 4, and the number of clustering seeds is 2 pixels. In addition, the number of Decision Trees is also set to 30.

### 3.5. Classification Accuracy Assessment

Classification accuracy is typically evaluated either through expert visual interpretation or by using a confusion matrix. The latter is widely used for land vegetation classification due to its objectivity and clarity in reflecting performance [50].

To avoid overestimation of classification accuracy due to randomness and overfitting, the study combines the confusion matrix method with k-fold cross-validation to evaluate the accuracy of the land use type classification results in the Ebinur Lake Basin. This study employs 5-fold cross-validation, where the training dataset is partitioned into 5 equally sized and non-overlapping subsets. These subsets collectively comprise the entire training dataset. In each iteration, one subset is held out for model accuracy evaluation. The final accuracy of the classification model is determined by averaging the accuracy metrics obtained from each iteration [51].

Assuming there are *X_i_* correctly classified pixels out of a total of *N* pixels, the overall accuracy (OA), Kappa coefficient (Kappa), producer’s accuracy (PA) and user’s accuracy (UA) evaluation index are calculated as follows:(7)$$OA=\frac{\sum_{i=1}^{k}X_{i}}{N}$$

Overall accuracy is the proportion of correctly classified pixels out of the total number of pixels [52,53].(8)$$Kappa=\frac{OA-P}{1-P},P=\frac{1}{N^{2}}(\sum_{i=1}^{c}X_{i}*\sum_{i=1}^{r}X_{i})$$

The Kappa coefficient refers to a measure of consistency of classification results [54,55].(9)$$PA=\frac{X_{i}}{\sum_{i=1}^{c}X_{i}}$$

The producer’s accuracy is the proportion of correctly classified pixels for a particular category relative to the total number of reference pixels for that category [56].(10)$$UA=\frac{X_{i}}{\sum_{i=1}^{r}X_{i}}$$

The user’s accuracy refers to the ratio of the number of correctly classified pixels in a specific category to the total number of pixels classified into that category [57].

## 4. Results

### 4.1. Composite Feature Construction and Optimization

To evaluate the relative importance of various feature variables, the Gini algorithm was applied to a dataset consisting of 48 features using the GEE platform in conjunction with feature space datasets. This approach enabled the calculation and subsequent ranking of feature importance values. To allow for consistent comparison across variables, the computed importance values were normalized [16]. Furthermore, the proportional contribution of each feature variable type to the overall variable ensemble was calculated, based on the established feature variable importance. This analysis aimed to elucidate the differential impacts of distinct feature variable types, as visually represented in Figure 6.

The relative importance of different feature variable types revealed that vegetation index features exhibited the highest influence, with an importance proportion of 39.1%. Band and texture features demonstrated moderately lower influence, accounting for 24.1% and 26.1%, respectively. Terrain features exerted the least influence, with a mere 10.7% contribution. However, when considering the importance of individual feature variables, Elevation and Slope within the terrain feature types showed relatively high proportions. Similarly, NDTI_Y, NDTI_M, NDVI_M, S3SI_Y, and S3SI_M dominated the vegetation index feature type; Blue_M, SWIR1_Y, and Green_Y were prominent within the band feature type; and Contrast_M, Var_M, and Contrast_Y exhibited high proportions within the texture feature type.

Building upon the acquired feature variable importance, an optimal feature selection was further achieved by integrating the OOBE. As depicted in Figure 7, the OOBE reached its minimum at the 23rd feature variable. Consequently, this method effectively reduced the 48 feature variables to 23, namely, Elevation, SLOPE, NDTI_Y, NDTI_M, NDVI_M, S3SI_Y, S3SI_M, Blue_M, RVI_M, NDWI_Y, MNDWI_M, SWIR1_Y, NDVI_Y, NDBI_Y, Green, Contrast, Var_M, NDBI_M, RVI_Y, Contrast, EVI_M, Savg_Y, and NIR_Y, thereby significantly mitigating data redundancy. Within the optimized feature set, the two most critical features were elevation-related, specifically, Elevation and SLOPE, while vegetation index features also constituted a substantial portion.

### 4.2. Analysis of Classification Results

In this study, optimized feature sets derived through feature variable selection processes within GEE environment were utilized to classify land use in the Ebinur Lake Basin for the years 2001, 2011, and 2021. Four classification algorithms were employed: Random Forest, Support Vector Machine, Gradient Boosting Decision Tree, and Object-Oriented Gradient Boosting Decision Tree. The classification results are visually depicted in Figure 8, and the corresponding quantitative evaluation metrics are summarized in Table 5.

In the land cover classification of the Ebinur Lake Basin, the RF classifier achieved a high overall accuracy of 93.56% and a Kappa coefficient of 92.46%. In contrast, the SVM classifier yielded a significantly lower overall accuracy of 81.47% and a corresponding Kappa coefficient of 78.25%, indicating a notable disadvantage compared to RF. While the GBDT and OO-GBDT classifiers demonstrated improved accuracy over SVM, the GBDT classifier attained an overall accuracy of 93.85% with a Kappa coefficient of 92.82%. The OO-GBDT classifier further enhanced the overall accuracy to 93.17%, while maintaining a Kappa coefficient of 92.03%. Overall, excluding the relatively poor performance of the SVM classifier, the RF, GBDT, and OO-GBDT models exhibited high classification accuracies.

However, cropland and artificial surfaces are significantly influenced by human activities, often manifesting as regular, block-like, or linear features in remote sensing imagery. The spectral and spatial differences between these features and their surrounding land cover types are typically distinct. While maintaining classification accuracy, employing an OO-GBDT classification model effectively mitigates the presence of small, fragmented patches. Furthermore, this approach refines the separation of narrow, elongated patches and smooths the boundaries of larger patches, resulting in a land use classification map with enhanced detail and accuracy. The application of this model provides a more effective approach for the investigation of land use classification and its spatiotemporal variations in the Ebinur Lake Basin.

Furthermore, the OO-GBDT classifier was employed to classify land use types in the Ebinur Lake Basin for the years 2011 and 2001, yielding satisfactory classification results (Figure 9). Specifically, the overall accuracy and Kappa coefficient for the 2011 classification were 88.93% and 86.96%, respectively, while for the 2001 classification, they were 84.76% and 81.84% (Table 6).

### 4.3. Spatial and Temporal Analysis of Regional Land Use Change

Spatiotemporal changes in different land use types in the Ebinur Lake Basin were analyzed based on land use classification results from 2001, 2011, and 2021, with particular attention to mutual conversions, area changes, and spatial pattern dynamics over the study period.

Multi-temporal land use maps of the Ebinur Lake Basin were analyzed. This determined the area and percentage of each land use type across various time periods (Table 7). Overall, Forest, Bare Land, and Cropland were the predominant land use types across the three periods. Grassland, Artificial Surface, Water Body, and Other land use types exhibited relatively limited distribution. Notably, during the period from 2001 to 2021, Woodland exhibited a gradual increase. Grassland trends shifted from a decrease to an increase. Conversely, Cultivated Land showed a reduction. These changes are primarily attributed to local policies implemented during this period, which strictly prohibited overgrazing and deforestation, as well as the influence of natural climatic variations.

Land use changes in the Ebinur Lake Basin were examined using a transition matrix (Figure 10) [58]. Over the past 20 years, significant land use type changes have occurred, primarily among Forest, Cropland, Artificial Surface, and Bare Land. Analysis of land cover changes revealed a shift of 852.74 km^2^ from Forest to Bare Land between 2001 and 2011. The magnitude of this transition increased in the following decade, with 1237.59 km^2^ of Forest converting to Bare Land from 2011 to 2021. Cropland dynamics are influenced by both anthropogenic and natural environmental factors. Between 2001 and 2011, Cropland experienced significant land cover conversions. Specifically, 572.91 km^2^ of Cropland transitioned to Forest, 743.67 km^2^ to Artificial Surface, and 713.91 km^2^ to Bare Land. The period from 2011 to 2021 evidenced a decline, with a corresponding 570.74 km^2^ shift towards Artificial Surface. Furthermore, between 2001 and 2011, approximately 654.39 km^2^ of Artificial Surface transitioned to Cropland, and 1133.7 km^2^ of Bare Land changed to Forest. Between 2011 and 2021, 491.38 km^2^ of Artificial Surface was converted to Cropland, and 520.91 km^2^ was converted to Bare Land. During the same period, 1069.51 km^2^ of Bare Land transitioned to Forest, and 696.34 km^2^ transitioned to Other.

To investigate the spatial distribution and temporal change patterns and direction of each land use type within the Ebinur Lake Basin, standard deviational ellipses were generated (Figure 11) [59]. Within the Ebinur Lake Basin over the recent 20 years, Grassland, Cropland and Bare Land underwent notable alterations in their spatial distribution. Changes for Forest, Artificial Surface, and Water Body were comparatively less significant. Forest distribution progressively shifted eastward during the first decade. However, the mean center of Forest showed minimal change in the second decade. Over the entire 20-year period, the spatial extent of Forest gradually expanded. During 2001–2021, the Grassland distribution showed an increasing southeast-to-northwest tilt. Its directionality became more distinct. The mean center correspondingly shifted northwest. During the first 10 years, Cropland’s spatial distribution changed significantly, observed as a gradual westward movement of its mean center. A gradual reduction in its spatial extent was noted over the 20-year period. Bare Land exhibited an increasing trend in spatial extent over the two decades, coupled with an eastward shift in its mean center. In contrast, Artificial Surface and Water Body land cover types displayed minimal changes in both spatial distribution and extent over the 20 years.

## 5. Discussion

Traditionally, researchers constructing land cover classification sample sets have relied on field surveys or high-resolution imagery interpretation. While this approach can ensure sample quality, it often lacks efficiency and is prone to subjectivity, resulting in inconsistencies between datasets created by different individuals [9,10]. Therefore, this study introduces an automated framework for sample generation. This method efficiently creates land use sample sets and enables their transfer across different years. Furthermore, the generated and transferred samples were evaluated, confirming the high quality (JM distance > 1.9) of the sample sets for all three time periods.

A multi-feature composite space was constructed using multi-temporal data, integrating characteristics from different land cover types. Features were derived from two sources: annual median composites and images from key phenological periods. The annual composites minimize seasonal noise and retain stable spectral properties. Images from key periods, processed using SAVI time-series analysis, effectively capture distinct phenological turning points for similar land cover types, improving dynamic feature expression and addressing the uncertainty caused by the subjective selection of data synthesis window nodes in previous studies [16]. Feature space optimization was performed using the Gini index and OOBE. This step aimed to reduce data redundancy and lessen the impact of irrelevant features on the subsequent classification.

Finally, this study employed a two-part strategy. First, we optimized the multi-feature composite feature space using a combined Gini coefficient and OOBE approach [43,44]. This method effectively reduced computational complexity and data redundancy. Compared to using a single method for feature optimization, the approach adopted in this study enhanced the robustness and stability of the feature optimization model. Second, we integrated simple non-iterative clustering with the GBDT classification model. This enhanced clustering efficiency and classification performance while ensuring accuracy. It also smoothed the land use classification boundaries and demonstrated significant noise reduction effects.

This study developed a framework for land use classification in the Ebinur Lake Basin. The framework integrates considerations of classification samples, features, and models. It effectively performs multi-temporal land use classification within the basin. This methodology can serve as a reference for applications in land use classification and target extraction. While the land use classification framework developed in this study demonstrated good applicability in the Ebinur Lake Basin, further validation across diverse study areas and scales is necessary to confirm its feasibility and utility. Additionally, certain uncertainties remain in the research process. Firstly, the generation of the target study area’s land use distribution relied on existing global land cover products, making the defined land use types dependent on the inherent classification system of these products. Secondly, although multi-year classification products and spatial filtering techniques were employed to mitigate the impact of existing product accuracy on automated sample generation, this influence could not be entirely eliminated. Finally, to analyze the long-term land use change characteristics in the Ebinur Lake Basin, this study prioritized temporal coverage over spatial resolution by using lower-resolution remote sensing data. Moreover, the selection of features inherently involved a certain level of subjectivity.

To address certain limitations inherent in the land use classification framework developed in this study, future research could consider the following: enhancing the temporal and spatial resolution of the remote sensing data; further evaluating the quality of automated classification sample generation; integrating multi-source data; incorporating vegetation phenological characteristics; and employing deep learning classification models.

## 6. Conclusions

In this study, we developed and implemented a land use spatiotemporal change analysis framework based on automated sample generation and multi-feature fusion, with the Ebinur Lake Basin as a demonstration area. Multi-source remote sensing data and existing classification products were integrated to generate and transfer training samples for the years 2001, 2011, and 2021. A multi-dimensional feature space comprising spectral, index-based, topographic, and textural features was constructed and subsequently optimized. By comparing and evaluating various machine learning classifiers, high-accuracy land use classification was achieved for the study area. Furthermore, based on the land use classification results from 2001, 2011, and 2021, the spatiotemporal changes of different land use types in the Ebinur Lake Basin were monitored. The main conclusions are as follows:Based on multi-temporal existing classification products, integrating spatial convolutional filters and spectral features enables the rapid and effective generation of high-quality, representative multi-year sample datasets. Using this approach, we generated 14,000 pixels for the Ebinur Lake Basin in 2021, 12,028 in 2011, and 10,997 in 2001, and the JM distance between samples across all three years exceeded 1.9.Based on the temporal reconstruction, a multi-dimensional feature space was constructed. To avoid data redundancy and potentially misclassified features, the Gini coefficient and OOBE algorithm were used to compress and optimize the feature space while maintaining classification accuracy, reducing the original 48 features to 23.Land use in the Ebinur Lake Basin for 2021 was classified using Random Forest, Support Vector Machine, Gradient Boosting Tree, and Object-Based Gradient Boosting Tree. The overall accuracies and Kappa coefficients were 93.56% and 92.46%, 81.47% and 78.25%, 93.85% and 92.82%, and 93.17% and 92.03%, respectively. Object-Based Gradient Boosting Tree effectively removed small patches while maintaining accuracy, resulting in a more detailed and accurate land use classification map. For 2011, the Object-Based Gradient Boosting Tree model yielded an overall accuracy of 88.93% and a Kappa of 86.96%. The 2001 classification using the same model achieved an overall accuracy of 84.76% and a Kappa of 81.84%.Based on multi-year land use maps of the Ebinur Lake Basin, we analyzed area changes, transfer matrices, and ellipse/centroid shifts for each land use type. Over the past two decades, Forest, Bare Land, and Cultivated Land were the dominant land use types. Significant land use transitions primarily occurred among Forest, Cultivated Land, Artificial Surfaces, And Bare Land. Notably, Grassland, Cultivated Land, And Bare Land within the Ebinur Lake Basin exhibited substantial changes in spatial distribution and patterns. Grassland distribution showed a trend tilting from southeast to northwest, with increased directionality and a corresponding average centroid shift to the northwest. Cultivated Land displayed a gradual westward shift of its average centroid. Bare Land showed an increasing spatial extent, coupled with an eastward movement of its average centroid.

In summary, this study provides a comprehensive framework for large-area land use type classification and spatiotemporal change analysis. It facilitates highly automated and accurate regional land use classification. The framework offers valuable technical references for automated sample generation, multi-feature fusion and optimization, and classification model refinement. Furthermore, it enables the effective acquisition of land use classification and spatiotemporal change information for other regions.

## Figures and Tables

**Figure 1 sensors-25-04314-f001:**
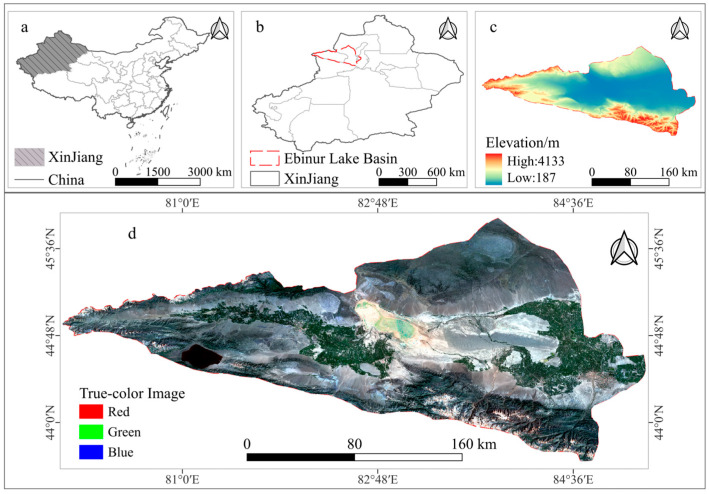
Location of the study area. (**a**) Xinjiang Uyghur Autonomous Region within China. (**b**) Location of the Ebinur Lake Basin. (**c**) Elevation Distribution of the Ebinur Lake Basin. (**d**) Landsat-8 image of the Ebinur Lake Basin.

**Figure 2 sensors-25-04314-f002:**
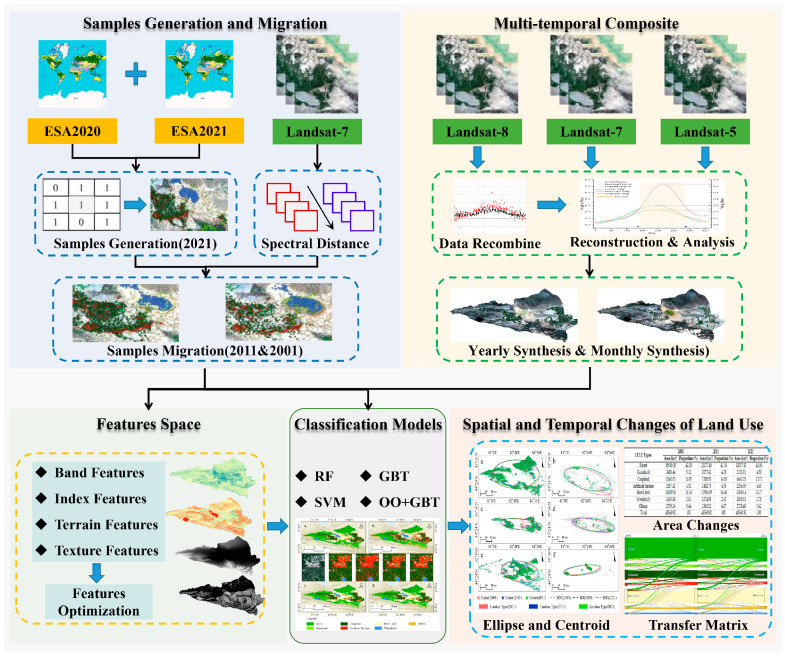
Workflow for land use type classification and spatiotemporal change analysis.

**Figure 3 sensors-25-04314-f003:**
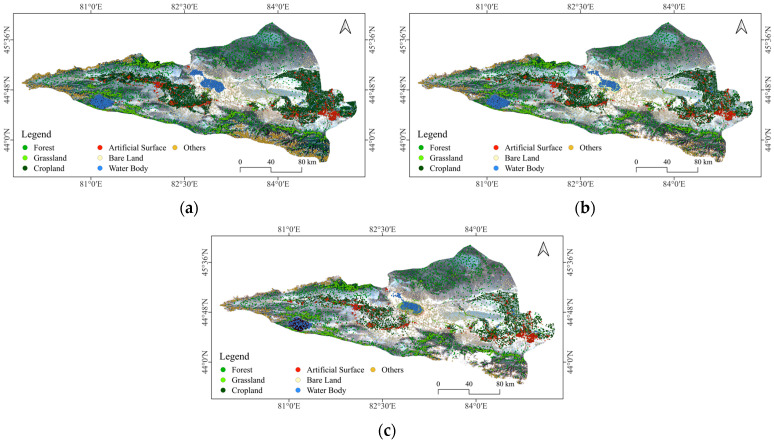
Land use classification samples distribution map. (**a**) Samples distribution of land use in 2021. (**b**) Samples distribution of land use in 2011. (**c**) Samples distribution of land use in 2001.

**Figure 4 sensors-25-04314-f004:**
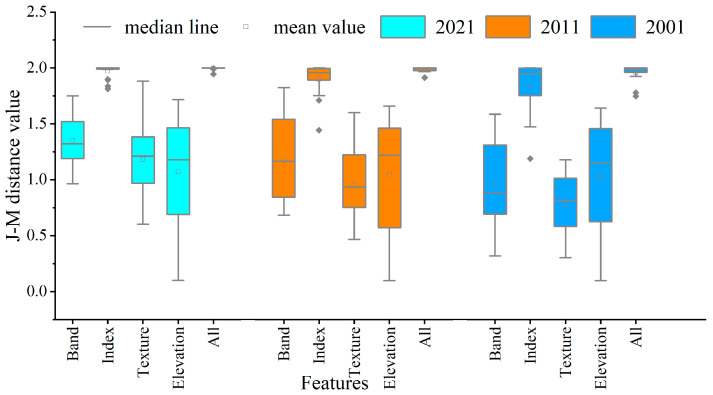
Separability of land use type sample sets in 2001, 2011, and 2021 across various feature types.

**Figure 5 sensors-25-04314-f005:**
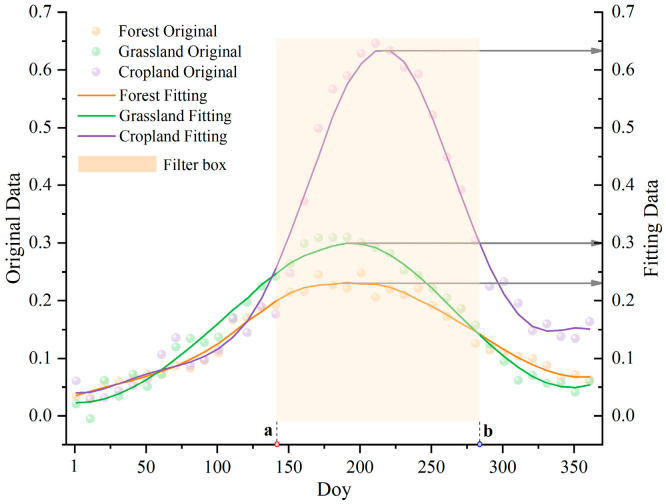
Time-series reconstruction and time-period filtering.

**Figure 6 sensors-25-04314-f006:**
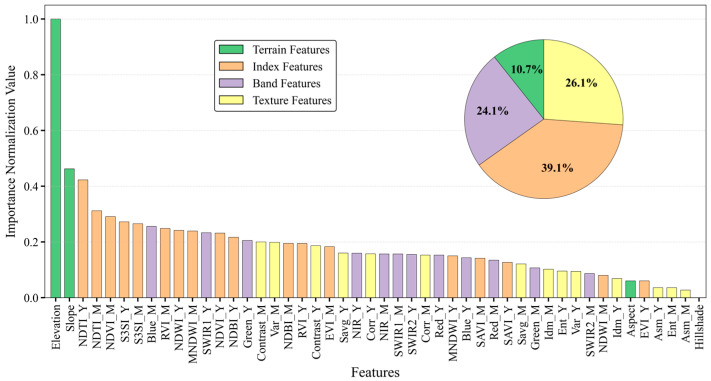
Feature importance percentage and sequence diagrams. The “Importance Normalization Value” refers to the degree of importance each feature holds in the classification process—the higher the value, the more important the feature.

**Figure 7 sensors-25-04314-f007:**
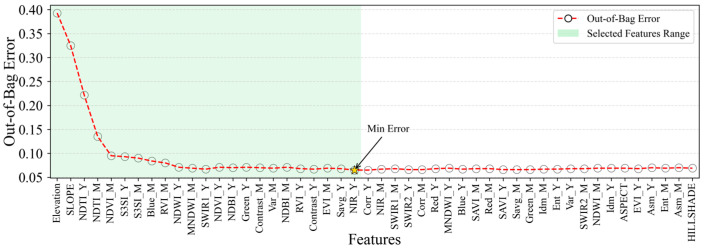
Features optimization and selection.

**Figure 8 sensors-25-04314-f008:**
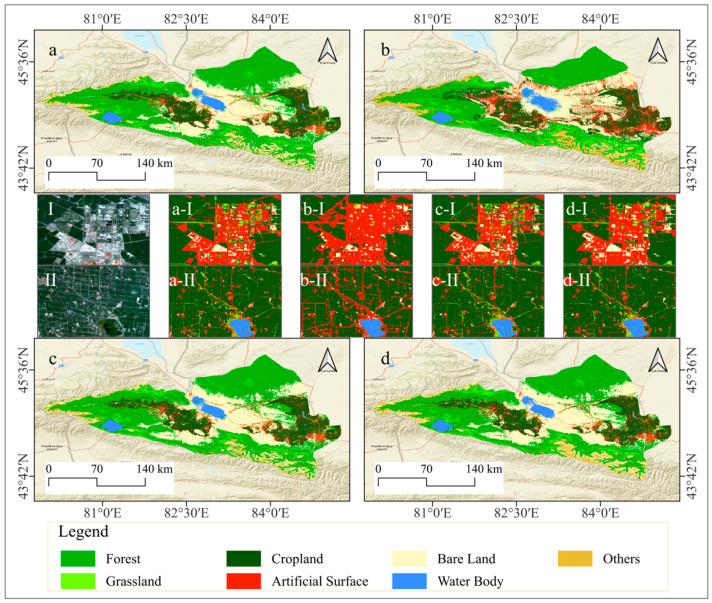
Multi-class model land use type mapping. (**a**) Random Forest classifier. (**b**) Support Vector Machine classifier. (**c**) Gradient Boosting Decision Tree classifier. (**d**) Object-oriented Gradient Boosting Decision Tree classifier. (**a-I**), (**a-II**), (**b-I**), (**b-II**), (**c-I**), (**c-II**), (**d-I**), and (**d-II**) show specific details in an area for comparison.

**Figure 9 sensors-25-04314-f009:**
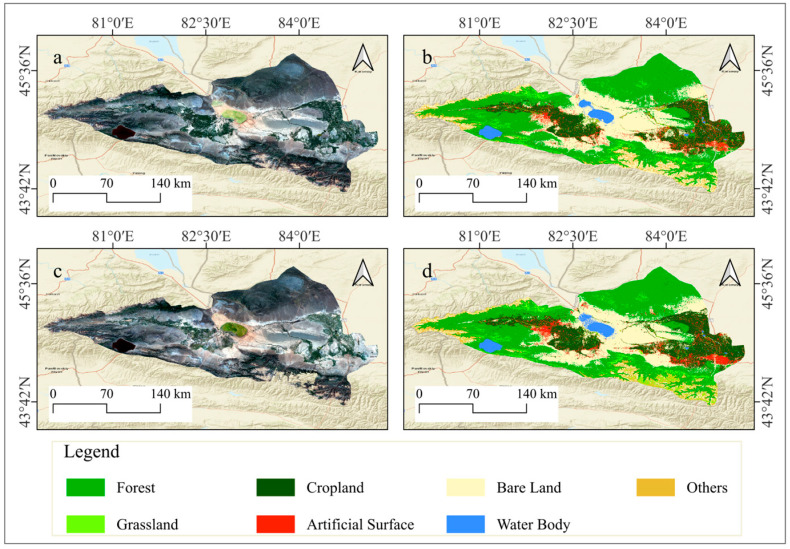
Land use classification maps of 2011 and 2001. (**a**) True-color imagery of the Ebinur Lake Basin in 2011. (**b**) Land use classification map of the Ebinur Lake Basin in 2011. (**c**) True-color imagery of the Ebinur Lake Basin in 2001. (**d**) Land use classification map of the Ebinur Lake Basin in 2001.

**Figure 10 sensors-25-04314-f010:**
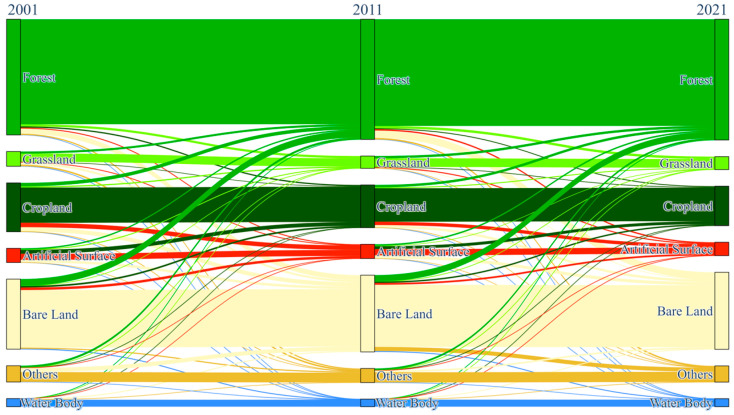
Changes in land use types in the Ebinur Lake Basin between 2001, 2011, and 2021.

**Figure 11 sensors-25-04314-f011:**
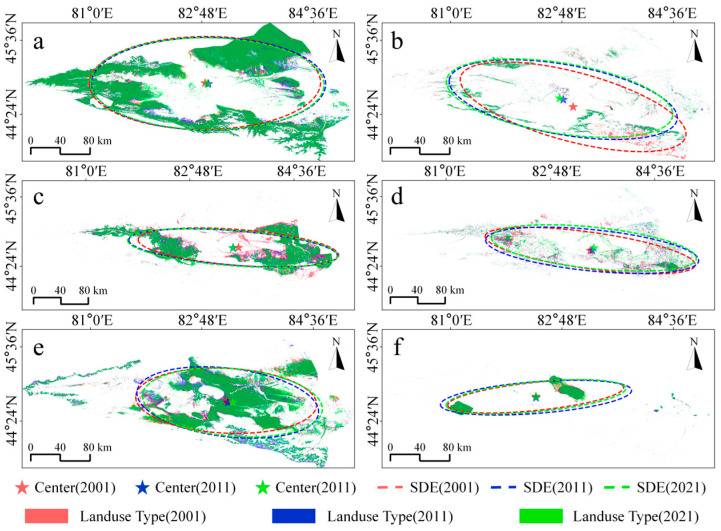
Standard deviation ellipse (SDE) and centroid (Center) changes for six land use types 2001, 2011, and 2021 in the Ebinur Lake Basin. (**a**) Forest. (**b**) Grassland. (**c**) Cropland. (**d**) Artificial Surface. (**e**) Bare Land. (**f**) Water Body.

**Table 1 sensors-25-04314-t001:** Landsat-5, Landsat-7 and Landsat-8 data band information (https://www.usgs.gov/faqs/what-are-band-designations-landsat-satellites (accessed on 15 February 2025)).

Satellite Sensor Name	Band	Wavelength (nm)	Resolution (m)
Landsat-5	Band 1-Blue	450–520	30
	Band 2-Green	520–600	30
	Band 3-Red	630–690	30
	Band 4-NIR	760–900	30
	Band 5-SWIR1	1550–1750	30
	Band 7-SWIR2	2080–2350	30
Landsat-7	Band 1-Blue	450–520	30
	Band 2-Green	520–600	30
	Band 3-Red	630–690	30
	Band 4-NIR	770–900	30
	Band 5-SWIR1	1550–1750	30
	Band 7-SWIR2	2090–2350	30
Landsat-8	Band 2-Blue	450–510	30
	Band 3-Green	530–590	30
	Band 4-Red	640–670	30
	Band 5-NIR	850–880	30
	Band 6-SWIR1	1570–1650	30
	Band 7-SWIR2	2110–2290	30

**Table 2 sensors-25-04314-t002:** The number of pixels for each land use classification in the years 2021, 2011, and 2021.

LULC Types	2021	2011	2001
Forest	2000	1973	1953
Grassland	2000	1961	1926
Cropland	2000	1999	1991
Artificial Surface	2000	1981	1977
Bare Land	2000	1828	1709
Water Body	2000	1457	767
Others	2000	829	674
Total	14,000	12,028	10,997

**Table 3 sensors-25-04314-t003:** Statistical parameters of S–G filter time-series reconstruction performance.

	Average Value	Standard Deviatior	Root Mean Square Error	Correlation Coefficient
Forest Original Data	0.13854	0.06766	\	\
Forest Filtering Data	0.13848	0.06613	0.01178	0.98433
Grassland Original Data	0.15859	0.09927	\	\
Grassland Filtering Data	0.15729	0.09618	0.01372	0.99046
Cropland Original Data	0.26911	0.20674	\	\
Cropland Filtering Data	0.26862	0.20057	0.02642	0.99181

**Table 4 sensors-25-04314-t004:** Features used for Classification.

Features Type	Parameter	Calculation Formula/Function
Band Features *	Blue	-
	Green	-
	Red	-
	NIR	-
	SWIR1	-
	SWIR2	-
Vegetation Index Features *	NDVI	(NIR−Red)/(NIR+Red)
	SAVI	(NIR−Red)/(NIR+Red+L)×(1+L),L=0.5
	EVI	2.5×(NIR−Red)/(NIR+6×Red−7.5×Blue+1)
	RVI	NIR/Red
	NDWI	(NIR−Green)/(NIR+Green)
	MNDWI	(Green−SWIR1)/(Green+SWIR1)
	NDTI	(SWIR1−SWIR2)/(SWIR1+SWIR2)
	NDBI	(SWIR2−NIR)/(SWIR2+NIR)
	S3SI	NIR×(Red−SWIR1)/(NIR+Red)×(NIR+SWIR1)
Texture Features *	Contrast	glcmTexture()
	Var	
	Savg	
	Corr	
	Idm	
	Ent	
	Asm	
Terrain Features	Elevation	-
	Aspect	ee.Terrain.aspect()
	Slope	ee.Terrain.slope()
	Hillshade	ee.Terrain.hillshade()

* The annual composite indices and the composite indices for the selected time periods are calculated using the median method. In the following text, they are denoted as “_Y” and “_M”, respectively.

**Table 5 sensors-25-04314-t005:** Multi-class model accuracy evaluation for land use classification.

LULC Types	RF		SVM		GBDT		OO-GBDT	
	PA(%)	UA(%)	PA(%)	UA(%)	PA(%)	UA(%)	PA(%)	UA(%)
Forest	92.44	90.77	74.45	79.60	91.18	92.61	90.96	90.71
Grassland	94.24	94.59	81.46	90.41	94.69	94.59	93.25	94.08
Cropland	96.89	97.70	86.46	79.34	96.95	97.73	96.60	96.35
Artificial Surface	93.91	92.20	86.42	75.58	94.87	91.69	92.97	91.09
Bare Land	89.92	88.79	57.71	70.43	89.40	90.52	87.74	89.58
Water Body	97.97	98.56	96.39	95.29	97.93	98.57	97.53	98.33
Others	87.88	91.97	89.65	78.76	91.18	90.21	93.09	92.14
OA(%)	93.56		81.47		93.85		93.17	
Kappa(%)	92.46		78.25		92.82		92.03	

**Table 6 sensors-25-04314-t006:** Accuracy statistics of land use classification in 2011 and 2001.

Classification Model		Forest	Grassland	Cropland	Artificial Surface	Bare Land	Water Body	Others	OA(%)	Kappa(%)
Class_2011 ^1^	PA(%)	89.64	89.45	87.60	85.04	86.65	98.76	86.38	88.93	86.96
	UA(%)	87.06	94.13	86.36	85.71	83.84	99.24	89.25		
Class_2001 ^2^	PA(%)	86.71	86.76	83.80	78.80	83.40	96.87	83.72	84.76	81.84
	UA(%)	87.50	92.22	75.29	82.24	81.57	97.62	88.61		

^1^ Land use type classification for the year 2011 using the OO-GBDT model. ^2^ Land use classification for the year 2001 using the OO-GBDT model.

**Table 7 sensors-25-04314-t007:** Land use area statistics for the years 2021, 2011, and 2001.

LULC Types	2001		2011		2021	
	Area (km^2^)	Proportion (%)	Area (km^2^)	Proportion (%)	Area (km^2^)	Proportion (%)
Forest	19,519.28	40.20	20,277.45	41.76	20,377.82	41.96
Grassland	2488.46	5.12	2077.42	4.28	2183.01	4.50
Cropland	8240.83	16.97	7208.93	14.85	6663.25	13.72
Artificial Surface	2387.41	4.92	2402.73	4.95	2256.95	4.65
Bare Land	11,819.81	24.34	12,954.39	26.68	13,001.4	26.77
Water Body	1365.88	2.81	1274.98	2.63	1350.01	2.78
Other	2739.26	5.64	2365.02	4.87	2728.48	5.62
Total	48,560.92	100	48,560.92	100	48,560.92	100

## Data Availability

All data generated or analyzed during this study are included in this article.

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
