# Peer review of "Spatiotemporal Land Use Change Detection Through Automated Sampling and Multi-Feature Composite Analysis: A Case Study of the Ebinur Lake Basin"

_sensors, 2025, doi:10.3390/s25144314_

Round 1

Reviewer 1 Report

Comments and Suggestions for Authors

The authors proposed the algorithm "Spatiotemporal Land Use Change Detection through Automated Sampling and Multi-Feature Composite Analysis: A Case Study of the Ebinur Lake Basin." After reviewing the manuscript, I have the following questions and comments that need to be addressed:

1.Throughout the paper, I did not find any actual results related to change detection; only classification results are presented. Should the title of the paper be redefined accordingly?

2.The procedures applied before data classification — can these be considered as data preprocessing? Please clarify.

3.In Section 3.4, the authors mention using RF, SVM, GTDB, and OO-GBDT algorithms. Relevant references for each method should be provided and listed in chronological order of publication.

4.In Section 4.3.5, references should also be added for each evaluation metric used. For example, the Kappa index can be cited using the following references:

[1] Li, L.; Ma, H.; Zhang, X.; Zhao, X.; Lv, M.; Jia, Z. Synthetic Aperture Radar Image Change Detection Based on Principal Component Analysis and Two-Level Clustering. Remote Sens. 2024, 16, 1861. https://doi.org/10.3390/rs16111861

[2] Cao, X.; Dong, M.; Liu, X.; Gong, J.; Zheng, H. Statistical Difference Representation-Based Transformer for Heterogeneous Change Detection. Sensors 2025, 25, 3740. https://doi.org/10.3390/s25123740

For OA (Overall Accuracy), the following references can be referenced:

[1] Qi, H.; Gao, X.; Lei, J.; Wang, F. IPACN: Information-Preserving Adaptive Convolutional Network for Remote Sensing Change Detection. Remote Sens. 2025, 17, 2121. https://doi.org/10.3390/rs17132121

[2] Li, J.; Zhang, H.; Chen, L.; He, B.; Chen, H. CSNet: A Remote Sensing Image Semantic Segmentation Network Based on Coordinate Attention and Skip Connections. Remote Sens. 2025, 17, 2048. https://doi.org/10.3390/rs17122048

5.Table 5 should maintain consistent decimal formatting to two decimal places. Additionally, the Kappa values should be presented as percentages.

6.The references should comply with the journal's formatting requirements. Please review and update them carefully.

Author Response

Response to reviewer:

Thanks for the valuable comments from you. We have revised our paper very carefully under those comments and suggestions that helped us a lot to improve the quality of this paper.

The following is our point-to-point response to your suggestions and questions, with the corresponding changes in the manuscript attached. (the relevant parts of the manuscript have been revised using the "highlight" function in word):

Reviewer Comments:

  1. Throughout the paper, I did not find any actual results related to change detection; only classification results are presented. Should the title of the paper be redefined accordingly?

Response:

Thank you to the reviewers for pointing out the issue of the title not aligning well with the main text. Firstly, the analysis of the spatiotemporal changes in land use in the Ebinur Lake Basin is based on the classification results from 2001,2011, and 2021. Secondly, in section 4.3, we focused on the transitions between different types of land use, changes in area, and spatial pattern shifts within the study period. Although we have rethought the title, considering the practical importance of monitoring spatiotemporal land changes in the study area, it is more appropriate to maintain the current title.

To clearly reflect this core research content, we have made the following key revisions to the manuscript: (1) In the abstract, we clearly state that the foundation of land use change detection and analysis is based on the results of three phases of land use classification, and we clearly outline the methods for change detection, specifically in lines 26-30; (2) At the beginning of section 4.3, we have added a clear statement emphasizing that "based on the land use classification results from 2001,2011, and 2021" specifically in lines 528-531; (3) In the conclusion, we have also added content on the change detection of land use based on classification results, specifically in lines 641-643. Additionally, in the final point of the conclusion, we have included research on the spatiotemporal changes in land use types in the Ebinur Lake Basin, specifically in lines 664-674.

  1. The procedures applied before data classification — can these be considered as data preprocessing? Please clarify.

Response:

In this paper, "data preprocessing" specifically refers to the simple and commonly used preprocessing operations applied to remote sensing imagery, such as band selection, cloud removal, and image clipping. However, the procedures conducted prior to data classification go beyond basic preprocessing of remote sensing images. They also include sample generation, multi-temporal data synthesis, and feature fusion, which are not merely preprocessing steps. Instead, these are subsequent research processes built upon the initial remote sensing image preprocessing and are generally not considered part of basic data preprocessing. Therefore, they should not be generalized in the same way as remote sensing image preprocessing.

In order to make this part more clear, we added the main content of this study point by point in Section 3, and isolated other operations before data preprocessing and classification, so as to distinguish the operations before data preprocessing and classification. The specific modification is shown in lines 177-181.

  1. In Section 3.4, the authors mention using RF, SVM, GTDB, and OO-GBDT algorithms. Relevant references for each method should be provided and listed in chronological order of publication.

Response:

In section 3.4, the references related to RF, SVM, GTDB, and OO-GBDT algorithms are added in chronological order to improve the organization and readability of the paper. The specific modification position is lines 373-377.

  1. In Section 4.3.5, references should also be added for each evaluation metric used. For example, the Kappa index can be cited using the following references:

[1] Li, L.; Ma, H.; Zhang, X.; Zhao, X.; Lv, M.; Jia, Z. Synthetic Aperture Radar Image Change Detection Based on Principal Component Analysis and Two-Level Clustering. Remote Sens. 2024, 16, 1861. https://doi.org/10.3390/rs16111861

[2] Cao, X.; Dong, M.; Liu, X.; Gong, J.; Zheng, H. Statistical Difference Representation-Based Transformer for Heterogeneous Change Detection. Sensors 2025, 25, 3740. https://doi.org/10.3390/s25123740

For OA (Overall Accuracy), the following references can be referenced:

[1] Qi, H.; Gao, X.; Lei, J.; Wang, F. IPACN: Information-Preserving Adaptive Convolutional Network for Remote Sensing Change Detection. Remote Sens. 2025, 17, 2121. https://doi.org/10.3390/rs17132121

[2] Li, J.; Zhang, H.; Chen, L.; He, B.; Chen, H. CSNet: A Remote Sensing Image Semantic Segmentation Network Based on Coordinate Attention and Skip Connections. Remote Sens. 2025, 17, 2048. https://doi.org/10.3390/rs17122048

Response:

Thank you very much for recommending several excellent articles. In our manuscript, we have cited relevant literature for each classification accuracy evaluation metric. The specific revisions can be found in lines 437–443.

  1. Table 5 should maintain consistent decimal formatting to two decimal places. Additionally, the Kappa values should be presented as percentages.

Response:

For Table 5 and the positions of classification accuracy evaluation indicators in the whole paper, two decimal places were retained uniformly, and the Kappa values was displayed in percentage form.

  1. The references should comply with the journal's formatting requirements. Please review and update them carefully.

Response:

By further referring to the format requirements of journal references, we have revised some of the problematic reference contents.

In addition, we further invited team doctoral members with overseas living experience to check the grammar, spelling and sentences in the text and make corresponding corrections. For the research status of related content, we have also combed it to some extent. Finally, thank you again for your useful suggestions. We believe this can improve the readability of the article. If you have any other questions or problems, we look forward to hearing from you. Thank you again for your time and consideration.

Reviewer 2 Report

Comments and Suggestions for Authors

A brief summary

Title: Spatiotemporal Land Use Change Detection through Automated Sampling and Multi-Feature Composite Analysis: A Case Study of the Ebinur Lake Basin.

This study aimed to automate sample generation, optimize multi-dimensional feature space composition, and refine classification models to achieve precise land use type classification and spatiotemporal change analysis in the Ebinur Lake Basin. This fully complies with the title of the paper.

The study area is the northwestern part of Xinjiang (China), on the southwestern fringe of the Junggar Basin (43°39′–45°53′N, 79°54′–85°16′E; 48561 km2). Elevation varies from 187 to 4133 m a.s.l.

In this study, the authors utilized multispectral Landsat imagery for 2001, 2011, and 2021. The data were processed using the Google Earth Engine (GEE) platform. Also, ESA classification products for 2020 and 2021 were applied to automate the generation of sample sets. The initial 48 features (bands, indexes, terrain, textures) were reduced to 23 using the Gini coefficient and out-of-bag error. These features were applied to generate classification maps using Random Forest, Support Vector Machine, Gradient Boosting Decision Tree, and a proposed object-oriented Gradient Boosting Decision Tree classification model.

The authors found that (1) The Object-Based Gradient Boosting Tree method effectively removed small patches while maintaining accuracy (Kappa = 0.87-0.92), resulting in a more detailed and accurate land use classification map; (2) Over the past two decades, forest, bare land, and cultivated land were the dominant land use types. Significant land use transitions primarily occurred among the forest, cultivated land, artificial surfaces, and bare land. Notably, grassland, cultivated land, and bare land exhibited substantial changes in spatial distribution and patterns. Grassland distribution showed a trend tilting from southeast to northwest, with increased directionality and a corresponding shift to the northwest. Cultivated land displayed a gradual westward shift. Bare land showed an increasing spatial extent, coupled with an eastward movement.

The manuscript is within the scope of the Sensors journal.

Broad comments

The Abstract conforms to the results and conclusions.

The Introduction is good enough, but there are no clearly stated aim of the research and questions to be solved.

The Study Area, Materials and Methods are described well enough but I suggest merging the “Study Area and Data Source” and the “Method” into the “Materials and Methods” section. Please, check the numbers of the tables.

The Results section conforms with the methods used.

The Discussion is sufficient but can be improved by comparing and adding references to similar investigations.

In my view, the manuscript requires minor revision.

The Conclusions agree with the results.

Specific comments

Introduction

L46: Should “and radar satellite/UAV data” be “radar satellite and UAV data”?

L112-127: I suggest rephrasing this part by the aim of the research and questions to be solved.

Materials and Methods

L214: Please explain “The thresholds are set at 0.4 and 0.5, respectively”. Do you mean a spectral angle and Euclidian distance? How it was used to enable the migration of the samples?

L225, Table 3: “The number of samples” or the number of pixels (L218)?

L244: In my view, it is better to use the “JM” abbreviation instead of “J-M” because of the absence of the minus sign.

L251-252: It is better to replace “J – M” with “JM”.

L276: Please explain what you mean by “Performance of land use type sample sets”. I only can see JMs in Figure 4.

L303: I suggest to use “–” instead of “\” to show empty cells in tables 4 and 2.

L335: Please check the number of the table. It should be number 5. Also, in the L325.

Results

L468: Please explain what means “_Y” and “_M” in the abbreviations of the features used. Annual and monthly data?

L465: Is “Elevation features” should be “Terrain features”?

L473: Please make the figure legend more self-explained. What are “Importance Normalization Values”? Is Elevation most important?

L562: Please add a space in “2011and”.

L567: Please add a space in “Waterbody”.

L581: Please add “SDE” to the legend.

L583: Please add a space in “Waterbody”.

Discussion

L585-624: Please add some references and comparisons with other investigations.  

Conclusions

L642: Are samples and pixels meaning the same?

References

69% of them are not older than 2019.

0% of them are self-citation.

Comments on the Quality of English Language

I'm not a native English speaker.

Author Response

Response to reviewer:

Thank you for your detailed, meticulous and valuable comments. We have revised our paper very carefully under those comments and suggestions that helped us a lot to improve the quality of this paper.

The following is our point-to-point response to your suggestions and questions, with the corresponding changes in the manuscript attached. (the relevant parts of the manuscript have been revised using the "highlight" function in word):

Reviewer Comments:

  1. Introduction
  • L46: Should “and radar satellite/UAV data” be “radar satellite and UAV data”?
  • L112-127: I suggest rephrasing this part by the aim of the research and questions to be solved.

Response:

  • Thank you very much for your correction. The revision has been made in line 46 of the manuscript.
  • Thank you for your suggestion. During this revision, we have rephrased this part based on the research objectives and the problems to be addressed. The specific modifications can be found in lines 112–119.

  1. Materials and Methods
  • L214: Please explain “The thresholds are set at 0.4 and 0.5, respectively”. Do you mean a spectral angle and Euclidian distance? How it was used to enable the migration of the samples?
  • L225, Table 3: “The number of samples” or the number of pixels (L218)?
  • L244: In my view, it is better to use the “JM” abbreviation instead of “J-M” because of the absence of the minus sign.
  • L251-252: It is better to replace “J – M” with “JM”.
  • L276: Please explain what you mean by “Performance of land use type sample sets”. I only can see JMs in Figure 4.
  • L303: I suggest to use “–” instead of “\” to show empty cells in tables 4 and 2.
  • L335: Please check the number of the table. It should be number 5. Also, in the L325.

Response:

  • In this study, the values 0.4 and 0.5 refer to the threshold values for the spectral angle and Euclidean distance, respectively. The core idea behind the implementation of sample transfer is to use these two metrics to measure the similarity of sample data across different years, thereby identifying and selecting unchanged samples. These unchanged samples can then be used for classification tasks in the target year, thus accomplishing the task of interannual transfer. In this manuscript revision, we have incorporated this sample transfer approach into the paper, specifically revised as shown in lines 207-211.
  • Thank you very much for your correction. The term “samples” has been changed to “pixels” in line 222of the ma
  • Thank you very much for your correction. The term "J-M" has been changed to "JM" throughout the manuscript.
  • Thank you very much for your correction. The term "J-M" has been changed to "JM" throughout the manuscript.
  • In this study, the term "performance of the land use type sample set" is intended to convey the separability of the samples. Generally, the better the separability of the samples, the more favorable it is for classification. Considering that readers may be confused by the original wording, we have revised it in this version to "separability of the land use type sample set" to enhance the clarity and readability of the paper.
  • Thank you very much for your correction. The symbol “-” in both tables has been replaced with “\”.
  • Thank you very much for your correction. We have checked and revised the numbering of all tables throughout the manuscript.

  1. Results
  • L468: Please explain what means “_Y” and “_M” in the abbreviations of the features used. Annual and monthly data?
  • L465: Is “Elevation features” should be “Terrain features”?
  • L473: Please make the figure legend more self-explained. What are “Importance Normalization Values”? Is Elevation most important?
  • L562: Please add a space in “2011and”.
  • L567: Please add a space in “Waterbody”.
  • L581: Please add “SDE” to the legend.
  • L583: Please add a space in “Waterbody”.

Response:

  • "_Y" and "_M" refer to the composite values calculated using the median compositing method for annual and selected time periods, respectively. In this revision, we have provided explanations for both in the note under Table 4, specifically in lines332-334.
  • Thank you very much for your correction. It has been revised accordingly, line 458.
  • In this study, the "normalized feature importance value" refers to the degree of importance each feature holds in the classification process—the higher the value, the more important the feature. These values have been normalized to facilitate visual comparison. For clarity, we have added annotations after the figure titles in this revision(lines466-468).
  • Thank you very much for your correction. It has been revised accordingly, line 560.
  • Thank you very much for your correction. The entire manuscript has now been revised accordingly, including the relevant content presented in the figures and tables.
  • Thank you very much for your correction.The abbreviations have now been added after the corresponding text in the figure titles to clearly indicate the relationship with the legends in thefigures11.
  • Thank you very much for your correction. The entire manuscript has now been revised accordingly, including the relevant content presented in the figures and tables.

  1. Discussion
  • L585-624: Please add some references and comparisons with other investigations.  

Response:

  • Thank you for your suggestion. We have now added comparative references in the Discussion section to enhance the credibility and scientific validity of the results.

  1. Conclusions
  • L642: Are samples and pixels meaning the same?

Response:

  • Yes, they mean the same thing, and they have now been unified as "pixels".

  1. References
  • 69% of them are not older than 2019.
  • 0% of them are self-citation.

Response:

According to the reviewers' comments, some references have been supplemented and revised.

We have reviewed and revised the article in light of Broad comments and Specific comments. In addition, we further invited team doctoral members with overseas living experience to check the grammar, spelling and sentences in the text and make corresponding corrections. For the research status of related content, we have also combed it to some extent. Finally, thank you again for your useful suggestions. We believe this can improve the readability of the article. If you have any other questions or problems, we look forward to hearing from you. Thank you again for your time and consideration.
